# RbX: Region-based explanations of prediction models

## Abstract

We introduce region-based explanations (RbX), a novel, model-agnostic method that uses only query access to quantify the sensitivity of the scalar predictions from a black-box model to local feature perturbations. RbX is based on a greedy algorithm for building a convex polytope that approximates a region of feature space where predictions are close to the prediction at some target point $x_0$. The geometry of the learned polytope — specifically the change in each coordinate necessary to escape the polytope — then explains the local importance of each feature in changing the model predictions. In particular, these "escape distances" can be standardized and ordered to rank the features by local importance. The RbX method is informed by a goal of detecting as many relevant features as possible in locally sparse prediction models, without including any features that do not enter in to the model. We provide a real data example and synthetic experiments to illustrate the encouraging performance of RbX in this respect.

## 1 Introduction

Suppose we have a prediction model $\hat{f}(x)$ to estimate a scalar outcome $y$ given a set of features $x \in \mathbb{R}^d$. We assume we do not have any knowledge about the functional form of $\hat{f}$. Rather, we consider $\hat{f}$ as a black-box function to which we only have query access, meaning we may compute the value of $\hat{f}(x)$ for any desired input $x$.

After making a prediction at a target point $x_0$, we seek to quantify the local importance of each feature on the prediction. Perturbations in certain features will be more influential than others in changing predictions from $\hat{f}$ near $x_0$. We would like a systematic way of identifying these features.

We distinguish our problem, which we call *local prediction importance*, from the questions of feature selection and feature importance. Feature selection methods, such as the LASSO for linear models (Tibshirani, 1996) and modern extensions like LassoNet for black-box models (Lemhadri et al., 2021), aim to select a small subset of features to generate a predictive model with greater accuracy and/or interpretability. In our setting, the prediction model $\hat{f}$ is fixed, and we seek only to faithfully explain the predictions of that model, without regard to the unknowable data-generating process that created the features and response. Feature importance methods include popular permutation-based approaches introduced by Breiman (2001) for random forests, which were extended to generic black-box models by Fisher et al. (2019) and to a local method by Casalicchio et al. (2018). These approaches also fix the prediction function $\hat{f}$, but provide importance measures based on changes in the predictive performance of $\hat{f}$ as various features are ignored, permuted, or otherwise perturbed. That is, they consider the extent to which changing features impacts the ability of $\hat{f}$ to approximate some ground truth function $f$. By contrast, the term *prediction importance* emphasizes the singular role of the numerical outputs of $\hat{f}$, setting aside how well these predictions approximate reality.

The distinction between local prediction importance and local feature importance is not always made in the literature. However, it is relevant for a user who only cares about understanding the output of a given black-box model, and does not want prediction explanations conflated with the underlying signal the model is trying to approximate.

Our proposed approach to local prediction importance is via region-based explanations (RbX). The method is "model-agnostic," meaning it does not use any knowledge about the structure of $\hat{f}$, relying

only on query access. We defer a detailed description of the algorithm to Section 3.1, but the main idea is to construct a polytope that approximates the region in feature space with prediction values "close" to the prediction at a target point $x_0$ (Section 2.1). We then argue that distances from $x_0$ to the boundaries of this polytope in directions parallel to the coordinate axes inform the local sensitivities of $\hat{f}$ to each feature in desirable ways according to our evaluation properties defined in Section 2.

## 1.1 PREVIOUS WORK

Existing approaches to local prediction importance can be broadly divided into two categories: surrogate methods and gradient-based methods. Surrogate methods locally approximate $\hat{f}$ by fitting a simpler prediction model that treats the predictions of $\hat{f}$ in a region near $x_0$ as the response. The weights assigned to each feature in this model are then used for local importance. For instance, LIME (Ribeiro et al., 2016) draws feature instances from a density centered at the target point $x_0$ and uses a linear surrogate. Lundberg & Lee (2017) propose Kernel SHAP (hereafter just SHAP), which they showed is an algorithmic approximation to fitting an additive surrogate model with weights corresponding to Shapley values.

Gradient-based methods consider infinitesimal regions on the decision surface and use the resulting first-order approximation to derive local feature importance. For example, Baehrens et al. (2010) provide local prediction importances based on the absolute value of the components of the gradient vector $\nabla \hat{f}(x_0)$; their approach for estimating this gradient is by fitting a global surrogate model using Parzen windows. Integrated gradients (Sundararajan et al., 2017) considers the line integral of the components of the gradient of $\hat{f}$ over a straight line path in feature space from a baseline point $x$ to $x_0$. Other gradient methods are not model-agnostic. For instance, DeepLIFT (Shrikumar et al., 2017) relies on backpropagation to estimate gradients in neural networks.

## 2 WHY REGION-BASED EXPLANATIONS?

In general, the "ground truth" local prediction explanation for a given model $\hat{f}$ and target point $x_0$ is ill-defined. While the explanations from procedures like SHAP and IG are derived based on some particular set of axioms, evidently there is not a consensus as to which axioms are more "desirable". Thus, we choose to develop RbX based on two less restrictive but likely less controversial properties which we call sparsity and detection power:

**Property 1.** *(Sparsity) A feature not involved in the prediction model $\hat{f}$ is assigned no importance.*

**Property 2.** *(Detection power) The locally relevant features for $\hat{f}$ are assigned highest importance.*

Sparsity requires that any feature that cannot change the predictions from $\hat{f}$ is assigned no importance. Of course, sparsity is not *sufficient* for a good local prediction importance method, though we view it as necessary. Conversely, a method that always assigns zero importance to every feature trivially satisfies sparsity, but fails to specify any potentially important features, hence the need for detection power. In designing RbX, we seek to maximize detection power while preserving sparsity.

Ideas similar to Property 1 and 2 are not new. Indeed they are used by the authors of LIME and L2X (Chen et al., 2018), a method that computes local feature scores by maximizing a variational relaxation of the mutual information between $y$ and the features $x$ encoded by a classifier $\hat{f}$, to evaluate their methods. For instance, the experiments in Ribeiro et al. (2016) show that LIME does a better job than some baseline methods in finding the features used in sparse logistic regression models and decision trees.

SHAP, IG, and other gradient methods satisfy sparsity axiomatically, yet LIME and L2X do not. While there is some subjectivity in the definition of locally relevant, we believe a reasonable sufficient condition for local relevance of feature $j$ is for the $j$-th component of $\nabla \hat{f}(x_0)$, the gradient of the prediction at $x_0$, to be nonzero. Then in the case that $\hat{f}$ is an additive regression model, Property 1 corresponds to assigning zero importance to all features with zero coefficients, while Property 2 means assigning nonzero importance to all other features. A simple gradient-based method using finite differences would then always perfectly satisfy both properties, as the set of features with nonzero gradients would always be precisely the relevant features. By contrast, LIME only does this 90%-92% of the time in the experiments from Ribeiro et al. (2016).

What remains is to improve detection power in nonlinear prediction models without sacrificing sparsity. In such models, a feature might be locally relevant near the target point $x_0$, even if $\nabla \hat{f}(x_0) = 0$. Sundararajan et al. (2017) motivate IG in this way, noting that gradient methods fail when $\hat{f}$ has zero gradient with respect to a particular feature, but still varies in that direction within a non-infinitesimal neighborhood that is considered locally relevant.

IG addresses this zero-gradient issue by accumulating gradients along the entire line segment between some baseline feature values $x$ and the target point $x_0$. However, this still only detects features with respect to which $\hat{f}$ varies infinitesimally somewhere along this line segment. There are a lot of additional areas of the feature space near $x_0$ where $\hat{f}$ could depend on a given feature. Our approach, RbX, examines the sensitivity of $\hat{f}$ in a large number of directions, while ensuring sparsity from a finite differences gradient method. It does so in a non-infinitesimal neighborhood of $x_0$, adapting the search space to cover a much larger portion of the region the user specifies as "close" to $x_0$, which we describe next.

## 2.1 CLOSE REGIONS

RbX asks the user to specify, for any given target point $x_0$, the values of predictions from $\hat{f}$ that are "close" to $\hat{f}(x_0)$, the prediction at the target point. There are often natural choices. For example, if $\hat{f}$ is a class probability from a binary classifier, the close region might contain all prediction values on the same side of the decision boundary. If $\hat{f}$ predicts a numeric medical outcome, and $x_0$ corresponds to a healthy patient, then the region might be the accepted range of healthy outcomes. For ease of exposition we assume the close region is an interval $\mathcal{I}$ of the form $[\hat{f}(x_0) - \epsilon_L, \hat{f}(x_0) + \epsilon_H]$. The interval $\mathcal{I}$ then depends on the user's choice of two nonnegative parameters $\epsilon = (\epsilon_L, \epsilon_H)$, which can vary with $x_0$. For instance, $\hat{f}$ predicts body mass index, for which the healthy range is 18-25, we might take $\epsilon_L = 4$ and $\epsilon_L = 3$ when the target patient satisfies $\hat{f}(x_0) = 22$, while for $\hat{f}(x_0) = 26$ we'd take $\epsilon_L = 1$ and $\epsilon_H = \infty$. The close region $\mathcal{E} = \{x \in \mathbb{R}^d \mid x \in \mathcal{I}\}$ will hereafter be referred to as the "$\epsilon$-close" region to emphasize the dependence on $\epsilon$. Points outside $\mathcal{E}$ are said to be "$\epsilon$-far" and the boundary of $\mathcal{E}$ is referred to as the "$\epsilon$-boundary."

We view the specification of closeness on the scale of the outcome as an advantage relative to methods like IG and SHAP which require specification of a "baseline" feature value $x$. The sum of all feature explanation scores from these methods at any target $x_0$ is constrained to equal to $\hat{f}(x_0) - \hat{f}(x)$. While Sundararajan et al. (2017) note that natural baselines exist in settings like image classification and sentiment analysis, for a general prediction or classification setting there may not be a canonical choice, and the feature attributions will be sensitive to the choice of baseline. SHAP's reliance on a baseline is eliminated by the cohort Shapley method of Mase et al. (2019), though cohort Shapley still retains an additivity constraint that all feature attributions must add up to $\hat{f}(x_0) - \bar{f}$, where $\bar{f}$ is the mean prediction on a set of $n$ observations. If $\bar{f}$ is not a meaningful value then the individual feature scores do not have a direct interpretation.

# 3 A POLYTOPE APPROXIMATION ALGORITHM

We are now ready to describe the RbX algorithm, which approximates the $\epsilon$-close region $\mathcal{E}$ by a polytope $\mathcal{P}$. It does so by approximating the $\epsilon$-boundary at various points by affine hyperplanes.

**Definition 1.** *A polytope $\mathcal{P} \subset \mathbb{R}^d$ is any finite intersection of affine halfspaces, i.e.*

$$\mathcal{P} \equiv \cap_{1 \leq k \leq K} H_k,$$

*where $H_k = \{x \in \mathbb{R}^d : x^T u_k \leq c_k\}$ is defined by its normal vector $u_k \in \mathbb{R}^d$ and intercept $c_k \in \mathbb{R}$.*

The use of a polytope approximation, as opposed to a smooth shape like an ellipsoid, enables sparsity.

## 3.1 A POLYTOPE APPROXIMATION ALGORITHM

RbX is a greedy procedure that constructs the polytope approximation $\mathcal{P}$ of the $\epsilon$-close region $\mathcal{E}$ one halfspace at a time (Algorithm 1). The algorithm requires a set of *context samples* $\mathcal{X} = \{x_i\}_{1 \leq i \leq n}$

that form the basis of the sampling procedure. These should be a dense collection of representative feature values, for instance from a possibly unlabeled training or validation set. To make the polytope $\mathcal{P}$ scale equivariant, all the features are scaled to have standard deviation 1 across the context samples. Only $\epsilon$-far context points are used by the remainder of the procedure.

---

**Algorithm 1** The RbX Algorithm

---

1: **Input:** target $x_0 \in \mathbb{R}^d$, closeness thresholds $\epsilon \succeq 0$, prediction model $\hat{f}$ with query access, context samples $\{x_i\}_{1 \leq i \leq n}$, maximum number of splits $K$.
2: Compute $s$, the vector of standard deviations of the $d$ features across the context samples $\{x_i\}_{1 \leq i \leq n}$.
3: Standardize $x_i \leftarrow \text{diag}(s)^{-1} x_i$ for $i = 0, 1, \ldots, n$.
4: **for** $i \in [1 : n]$ if $x_i$ is $\epsilon$-far **do**
5:     Shrink $x_i$ onto the $\epsilon$-decision boundary using line search: $\tilde{x}_i \leftarrow \text{Line-Search}(x_i, \hat{f}, \epsilon, x_0)$
6: **end for**
7: Initialize $\mathcal{R} \leftarrow \{\tilde{x}_i\}$
8: Initialize the set of support vectors $\mathcal{S} \leftarrow \emptyset$
9: Initialize $k \leftarrow 1$
10: **while** $\mathcal{R} \neq \emptyset$ and $k \leq K$ **do**
11:     Find $\tilde{x}^{(k)} \leftarrow \underset{\tilde{x} \in \mathcal{R}}{\arg\min} \|\tilde{x} - x_0\|_2^2$
12:     Estimate gradient of $\hat{f}$ at $\tilde{x}^{(k)}$: $g_k \leftarrow \text{Estimate-Grad}(\tilde{x}^{(k)}; \hat{f}, \delta, r, m)$
13:     Compute halfspace: $H_k \leftarrow \{x \in \mathbb{R}^d \mid x^T g_k \leq (\tilde{x}^{(k)})^T g_k\}$
14:     Update region: $\mathcal{R} \leftarrow \mathcal{R} \cap \text{int } H_k$
15:     Update support vectors: $\mathcal{S} \leftarrow \mathcal{S} \cup \{\tilde{x}^{(k)}\}$
16:     Increment $k$: $k \leftarrow k + 1$
17: **end while**
18: **return** $\{H_k\}_{1 \leq k \leq n}$, the collection of halfspaces defining the polytope $\mathcal{P}$
19: Notation: Line-Search is defined in Algorithm 3. Estimate-Grad is given in Algorithm 2.

---

The first step in the RbX algorithm is to shrink each (standardized and $\epsilon$-far) context point $x_i$ along the line segment in $\mathbb{R}^d$ between $x_i$ and $x_0$ to a point on the $\epsilon$-boundary. This can be done quickly to exponential accuracy via a standard bisection-based line search (see Algorithm 3 in the Appendix for an example), noting that $x_0$ is always $\epsilon$-close.

Next, RbX finds $x^{(1)}$, the context point whose shrunken counterpart $\tilde{x}^{(1)}$ is closest to $x_0$ in Euclidean distance. The first halfspace $H_1$ of the polytope $\mathcal{P}$ is chosen to pass through $\tilde{x}^{(1)}$ and have normal vector equal to an estimate of the gradient of $\hat{f}$, computed using finite differences (Algorithm 2) at $\tilde{x}^{(1)}$. This is motivated by the fact that in the case that $\hat{f}$ is differentiable, $H_1$ is a first-order approximation of a level set of $\hat{f}$. Finally, all shrunken context points outside the interior of $H_1$ are discarded, and the process is iterated with the remaining shrunken context points until either $K$ halfspaces have been learned, or there are no more shrunken context points remaining.

Note that $\tilde{x}^{(k)}$, the shrunken context point chosen on the $k$-th iteration of the algorithm, lies outside the interior of the $k$-th halfspace $H_k$ by construction. Thus the number of remaining shrunken context points decreases by at least one after each iteration, and so the algorithm terminates in at most $n$ iterations. In practice, far fewer iterations are often needed. To further reduce computation, one can impose early stopping by specifying a maximum number of halfspaces $K < n$ to be learned.

The "greedy" nature of the RbX algorithm refers to how it chooses the closest shrunken context points first. This helps enforce a better approximation of the parts of the $\epsilon$-decision boundary closer to $x_0$. In general, any given shrunken context point may not be the *closest* point to $x_0$ that lies on the $\epsilon$-boundary and also along the ray from $x_0$ to the original context point. Thus, the algorithm may miss parts of the $\epsilon$-boundary extremely close to $\hat{f}$. But such an issue is mitigated by having a sufficiently dense set of context samples covering the full range of plausible feature values. Then the greediness ensures the closer parts of the decision boundary — among those corresponding to plausible feature values, where we are most interested in the predictions — are prioritized.

### 3.2 FINITE DIFFERENCES GRADIENT ESTIMATION AND SPARSITY

The gradient of $\hat{f}$ is estimated at each shrunken context point using finite differences (Algorithm 2). We allow the user to average gradient estimates at $m$ "jittered" points that are the original points corrupted by a small amount of Gaussian noise. This jittering is designed to smooth the gradient estimate when $\hat{f}$ has discontinuities on the $\epsilon$-boundary. If $\hat{f}$ has no dependence on the $i$-th feature, it is clear that the $i$-th component of $\nabla \hat{f}(x)$ will be 0 according to Algorithm 2, regardless of the parameters used. Note this would not be the case if Algorithm 2 were replaced by a smooth gradient estimator, such as the Parzen windows used by Baehrens et al. (2010). Having zero gradient estimates for irrelevant features ensures that our local importance scores satisfy sparsity, as described in Section 4. All results in this paper are presented with parameters $r = 0.01$, $\delta = 0.1$, and $m = 10$.

---

**Algorithm 2** Estimate-Grad Algorithm - Finite Differences

---

1: **Input:** point $x \in \mathbb{R}^d$, prediction model $\hat{f}$, step size $\delta$, jitter radius $r$, number of jitter samples $m$
2: **for** $j \in [1:m]$ **do**
3:      Generate $z \sim r \cdot \mathcal{N}(0, I_d)$
4:      $v \leftarrow x + z$
5:      $g_i^{(j)} \leftarrow \frac{\hat{f}(v+\delta e_i) - \hat{f}(v - \delta e_i)}{2\delta}, i = 1, \ldots, d$
6: **end for**
7: **return** $\nabla \hat{f}(x) = \frac{1}{m} \sum_{j=1}^{m} (g_1^{(j)}, ..., g_d^{(j)})$

---

### 3.3 TOY EXAMPLE

We briefly illustrate the RbX algorithm in Fig. 1, in a toy example with $d = 2$, $\hat{f}(x) = x_1 \cdot x_2$, $x_0 = (0, 0)$, and $\epsilon = (0.5, 0.5)$. 500 context points generated from a standard bivariate Gaussian distribution were used.

Each iteration of the algorithm approximates a plane tangent to the $\epsilon$-boundary at the closest shrunken context point. All context points on the side of the plane not containing $x_0$ are then discarded. The resulting $\mathcal{P}$ at termination (after 4 splits) is a diamond-shaped region that truncates the true $\epsilon$-close region, which is non-convex and extends infinitely along the coordinate axes in both directions.



Figure 1: An illustration of the RbX algorithm for the pairwise interaction model $\hat{f}(x) = x_1 \cdot x_2$ with $\epsilon = (0.5, 0.5)$ and target point $x_0 = (0, 0)$, highlighted in the center of each plot. The smaller dots are the context points, colored by whether they are $\epsilon$-far. Each panel shows the additional halfspace constructed in one iteration of the RbX algorithm. The lines are the halfspaces learned at each step of the algorithm; the larger dots along these lines are the shrunken context points used.

## 4 FROM POLYTOPES TO LOCAL PREDICTION IMPORTANCE

Given the polytope $\mathcal{P}$ output by the RbX algorithm, we derive local prediction importance scores $S_j(\mathcal{P})$ using "feature escape distances" for each feature $j \in \{1, \ldots, d\}$:

$$S_j^+(\mathcal{P}) = \inf\{\alpha > 0 \mid x_0 + \alpha e_j \notin \mathcal{P}\}; \quad S_j^-(\mathcal{P}) = \inf\{\alpha > 0 \mid x_0 - \alpha e_j \notin \mathcal{P}\};$$

$$S_j(\mathcal{P}) = \min(S_j^+(\mathcal{P}), S_j^-(\mathcal{P})) \cdot \text{sign}(S_j^+(\mathcal{P}) - S_j^-(\mathcal{P}))$$

Here $e_j$ is the $j$-th standard basis vector in $\mathbb{R}^d$. Then $S_j(\mathcal{P})$ is the minimum signed distance needed to escape the RbX polytope $\mathcal{P}$ by varying only the $j$-th feature from the target point $x_0$. As argued by Mase et al. (2019), predictions from $\hat{f}$ at feature values that are implausible should likely not be trusted. Thus, whenever the "escape path" out of the polytope $\mathcal{P}$ in the direction of feature $j$ passes outside the region $\mathcal{T}$ where predictions from $\hat{f}$ are "trustworthy," we set $S_j(\mathcal{P})$ to $\infty$. Appendix A.2 provides some methods and examples to determine how to construct $\mathcal{T}$ based on the context points.

Note that the escape distances $S_j(\mathcal{P})$ are reported on the original scales of each feature (before standardization in step 3 of Algorithm 1), which enables them to be interpreted individually without reference to the escape distances of the other features. Alternatively, we can report the escape distances $\tilde{S}_j(\mathcal{P})$ on the standardized scale. Sorting these standardized distances (from smallest to largest) enables us to obtain a ranking of the local importance of the features (from most important to least important).

If $\hat{f}$ doesn't depend on a feature $x_j$, then all of the halfspaces defining $\mathcal{P}$ will have normal vectors with 0 component in the $x_j$ direction. This implies that the corresponding $S_j(\mathcal{P})$ will be $\infty$, and thus such features will have no importance, showing our procedure satisfies sparsity.

It is natural to compare the feature escape distances $S_j(\mathcal{P})$ with the "simple feature escape distances" $S_j(\mathcal{E})$, which use the original $\epsilon$-close region $\mathcal{E}$ in place of the polytope $\mathcal{P}$. They can be computed via a line search similar to Algorithm 3, without running RbX. Clearly, a feature ranking based on the $S_j(\mathcal{E})$ would also satisfy sparsity. Furthermore, it would have better detection power than a gradient-based method. This is because the $S_j(\mathcal{E})$ look beyond an infinitesimal neighborhood of $x_0$, instead focusing on a typically larger region defined in terms of the prediction values of $\hat{f}$ to be meaningful. However, it still cannot capture changes in $\hat{f}$ near $x_0$ that cannot be detected when only one feature is changed at a time from its value at $x_0$.

Since the polytope $\mathcal{P}$ looks in many directions around $x_0$, using the RbX distances $S_j(\mathcal{P})$ provides better detection power. A simple example of this can be seen in Fig. 1. For $x_0 = (0, 0)$ we have $S_1(\mathcal{E}) = S_2(\mathcal{E}) = \infty$ since all points along the coordinate axes are $\epsilon$-far. Yet $S_1(\mathcal{P}) \approx S_2(\mathcal{P}) \approx 1.4$, giving the two features equal and nonzero importance. Further examples are in the next section.

## 5 DATA EXAMPLE AND SYNTHETIC EXPERIMENTS

We begin by comparing RbX to existing methods for local prediction importance on a credit scoring example, considering sparsity, detection power, and robustness across similar target points. Then we consider simulated experiments designed to more systematically evaluate detection power.

### 5.1 CREDIT SCORING EXAMPLE

The home equity line of credit (HELOC) dataset from the FICO xML Challenge (`community.fico.com/s/xml`) contains the `RiskPerformance` of 2,502 credit applicants. This is a binary indicator of whether they were ever more than 90 days past due in the first two years after account opening. The goal is to interpretably classify each individual as having either "Bad" or "Good" `RiskPerformance` based on 23 quantitative or ordered categorical predictors. We consider the local prediction importance of a shallow and sparse decision tree. In Appendix A.3 we consider a sparse gradient boosted tree ensemble fit to the same data. After splitting the dataset randomly into 1,751 training observations and 751 test observations, the tree classifier is fit to the training observations to have depth 3. The final classifier $\hat{f}$ is visualized in Fig. 2. The scalar prediction output is taken to be the predicted probability of a "Good" `RiskPerformance`.

We begin by examining a specific target point, corresponding to the individual labeled 5,238 in the dataset. Selected feature values for that individual (who has prediction value 0.191) are given in Table 1, along with feature scores from various local importance methods. For LIME and SHAP, we compute the scores from the open-source implementations and default hyperparameter settings, except that for LIME we do not discretize the features, which greatly improves its performance. For RbX we assume a decision boundary of $\{x \mid \hat{f}(x) = 0.5\}$, and take $\mathcal{E}$ to be the set of points on the same side of this decision boundary as $x_0$. All training observations are used as context points.

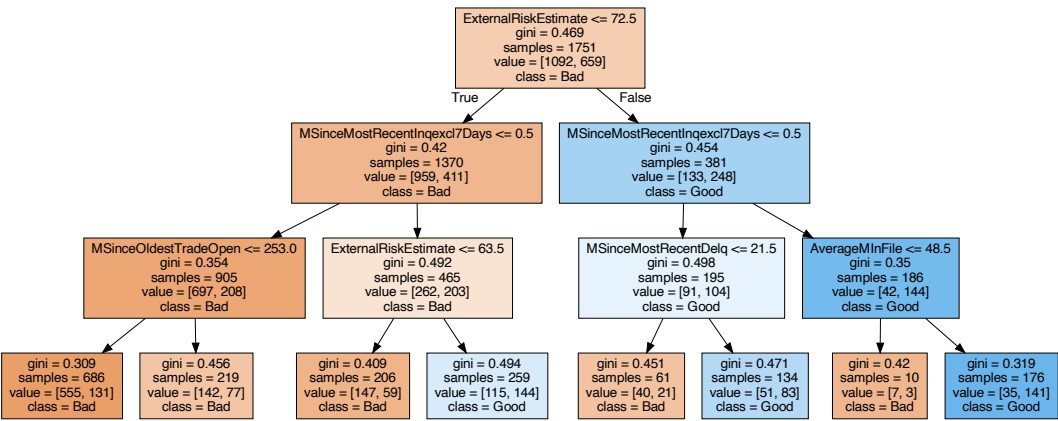

Figure 2: A visualization of a classification tree fit to the HELOC data from the FICO xML Challenge.

Table 1: Feature values and various local prediction importance scores for the sample target point analyzed in the main text, with the prediction model $\hat{f}$ as in Fig. 2. Note the scores from LIME and SHAP (for all features) are on the scale of the outcome, whereas the unnormalized escape distances $S_j(\mathcal{P})$ from RbX are on the scale of each feature $j$.

| Feature | Feature value | $S_j(\mathcal{P})$ | $\tilde{S}_j(\mathcal{P})$ | LIME | SHAP |
|---|---|---|---|---|---|
| ExternalRiskEstimate | 61.0 | 2.8 | 0.37 | 0.12 | -0.13 |
| MSinceOldestTradeOpen | 149.0 | $\infty$ | $\infty$ | 0.018 | 0 |
| MSinceMostRecentInqexcl7Days | 0.0 | 0.5 | 0.12 | 0.084 | -0.23 |
| MSinceMostRecentDelq | 3.0 | $\infty$ | $\infty$ | 0.0067 | 0 |
| AverageMInFile | 49.0 | $\infty$ | $\infty$ | 0.016 | 0 |
| NumTrades60Ever2DerogPubRec | 1 | $\infty$ | $\infty$ | 0.0011 | 0 |

The gains to detection power provided by the polytope $\mathcal{P}$ constructed by the RbX algorithm is seen by noting that the simple feature escape distances $S_j(\mathcal{E})$ are infinite for *every* feature, as changing any single feature in the target point — and keeping all others fixed — cannot change the target individual's classification from Bad to Good under the tree model. Thus simple feature importance is fully uninformative. By contrast, we have finite feature escape distances $S_j(\mathcal{P})$ of 2.8 for `ExternalRiskEstimate` and 0.5 for `MSinceMostRecentInqexcl7Days`. This is sensible, as we can see from Fig. 2 that increasing both of these features simultaneously by these two amounts is close to the shortest path for the individual to change their classification to `Good`.

Next, we evaluate the stability of the various local importance methods across parameter choices. We consider 100 randomly chosen target points from the training data. At each target point we run 259 iterations of SHAP, LIME, and RbX. Each SHAP iteration corresponds to a different choice of baseline among those points in the leaf of the tree classifier with gini impurity score 0.494 (Fig. 2). One such baseline is used for Table 1. Each LIME iteration corresponds to an independent run of the LIME sampling procedure, and each RbX iteration uses a different set of context samples. Each set of context samples is generated by adding independent Gaussian noise to each feature dimension of the points in the training set. The standard deviation of this noise is taken to be the (marginal) sample standard deviation of the relevant dimension across the 259 baselines used for SHAP.

We consider the standard deviation of the ranks of the 5 features that enter into the tree classifier across iterations for each target point. The rank of any feature assigned zero importance is defined to be one greater than the number of features with nonzero importance. We then average these standard deviations across target points and the dimensions. The result is 0.225 for SHAP, 0.193 for RbX, and

0.316 for LIME, demonstrating RbX predictions are quite stable with respect to the choice of context samples. We also note that for LIME, one of the 18 features not entering into the tree classifier at all is assigned greater importance than one of the other 5 features in 7.9% of all iterations across the target points. Note this cannot occur for SHAP and RbX thanks to sparsity.

## 5.2 DETECTION POWER EXPERIMENTS

We now evaluate the detection power of various local prediction importance methods on locally sparse models, where the "ground truth" set of locally relevant features is clear. One closely related evaluation framework would be that proposed by Zhou et al. (2022), which involves cleverly introducing noise into a training dataset and then fitting various non-sparse prediction models to such a dataset. If these models have sufficiently high performance, then it is clear they must place primary importance on the noisy features. They evaluate (global) feature attribution methods based on the relative importance of such features. While such a method enables measuring the performance of more complex, truly black-box models that might be used in high signal-to-noise applications like image classification, it is not necessary for the simpler models examined here.

We consider the following four synthetic data generating scenarios, inspired by Chen et al. (2018):

1. Generate $(X_1, \ldots, X_9)$ from a spherical standard Gaussian distribution. Generate $X_{10}$ independently from an equally weighted mixture of two Gaussian distributions with standard deviation 1, centered at $+3$ and $-3$.
2. Let $X = (X_1, \ldots, X_{10})$. Then an outcome $Y$ is generated as follows:
   - XOR: $\mathbb{E}(Y \mid X = x) = (1 + x_1 x_2)^{-1} = p_X(x_1, x_2)$
   - Orange skin: $\mathbb{E}(Y \mid X = x) = \left(1 + \exp\left(\sum_{i=1}^{4} x_i^2 - 4\right)\right)^{-1} = p_O(x_1, x_2, x_3, x_4)$
   - Nonlinear additive: $\mathbb{E}(Y \mid X = x) = (1 + \exp(-100 \sin(2x_1) + 2|x_2| + x_3 + \exp(-x_4)))^{-1} = p_N(x_1, x_2, x_3, x_4)$
   - Feature switching: $\mathbb{E}(Y \mid X = x) = p_O(x_1, \ldots, x_4) r(x_{10}) + p_N(x_5, \ldots, x_8)(1 - r(x_{10}))$

   where $r(x_{10}) = \exp\left(-\frac{(X_{10}-3)^2}{2}\right) / \left(\exp\left(-\frac{(X_{10}-3)^2}{2}\right) + \exp\left(-\frac{(X_{10}+3)^2}{2}\right)\right)$

Feature switching can be conceptualized as the following method of generating $Y$ given $X$. If $X_{10}$ is drawn from the Gaussian component with center $+3$, $Y$ is generated according to the orange skin process with features $X_1, \ldots, X_4$. Otherwise, $Y$ is generated according to the nonlinear additive process with features $X_5, \ldots, X_8$. The function $r(x_{10})$ is the posterior probability that $X_{10}$ was drawn from the component with center $+3$, given $X_{10} = x_{10}$.

For each scenario we consider two prediction models — the Bayes prediction model $\hat{f}(x) = \mathbb{E}(Y \mid X = x)$ and a K-nearest neighbors (KNN) regressor. The KNN regressor for each scenario is fit to a random sample of 1,000 training points with $K = 5$, considering only the locally relevant features. For feature switching, features 1-4 and 10 are considered locally relevant for a target point with $x_{10} \geq 0$; otherwise, features 5-8 and 10 are locally relevant. In the other scenarios, the locally relevant features for all target points are the ones that can affect predictions in the Bayes classifier, e.g. 1-4 for the orange skin and nonlinear additive settings.

We evaluate the detection power of each local importance method across 1,000 randomly chosen target points by computing the average proportion of locally relevant features recalled among the top $M$ most important features. Here $M$ is the true number of locally relevant features. Ties in importance scores are broken randomly but features with zero importance are never selected. We compare the RbX escape distances $S_j(\mathcal{P})$ with the local importance scores from the following methods, which have well-documented public implementations or are trivial to code: simple feature importance (SFI), a gradient method using Algorithm 2 ("Gradient"), IG, LIME, and SHAP. For RbX we use 1,000 independently sampled context points. Only LIME does not satisfy sparsity among the approaches considered. The $\epsilon$-close region for each target point consists of all points on the same side of the decision boundary $\{x \mid \hat{f}(x) = 0.5\}$. For SHAP the baseline point is the origin.

For all scenarios, RbX shows the highest recall among the methods considered, as the only procedure with perfect performance on both classifiers for all scenarios besides feature switching (Fig. 3).

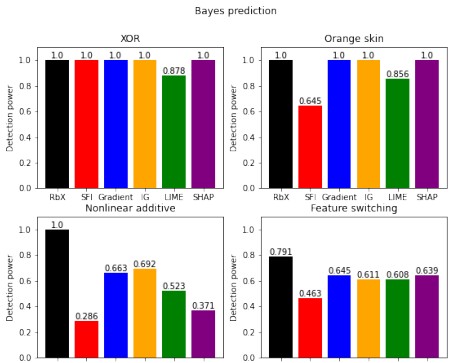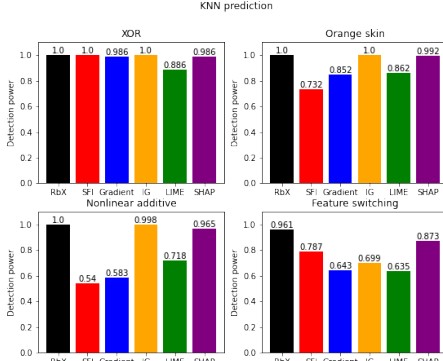

Figure 3: (Left) Recovery rates, based on 1,000 random target points, of the locally relevant features for various local prediction importance methods, using the smooth, sparse Bayes prediction model for each of the 4 simulated scenarios described in the main text. (Right) Same as the left panel, but for the nonsmooth KNN prediction model.

The gradient-based and simple feature importance methods particularly struggle with the nonlinear additive model since there are regions where the classifiers are locally flat — for instance, wherever $\sin(2X_1)$ is sufficiently greater than zero so that the predictions in a neighborhood are all numerically equivalent to 1. RbX overcomes this by examining a non-infinitesimal neighborhood around $x_0$.

## 6 DISCUSSION, LIMITATIONS, AND FUTURE WORK

We have proposed region-based explanations (RbX) as a novel, model-agnostic approach for local prediction importance. The idea is to successively refine polytope approximations to a region of feature space with similar predictions to the target point. The user can directly specify what prediction values are "similar" based on context. By contrast, existing methods specify a "locally relevant" region in terms of the values of the *features*. When the number of features is moderate and there are interactions between them, specifying such a region becomes difficult. We've argued that RbX a strong ability to detect locally relevant features compared to existing methods while preserving sparsity, i.e. a guarantee that features not used for prediction are assigned zero importance.

Whereas many local prediction importance methods like LIME and SHAP were designed with binary or unordered categorical features in mind, RbX is motivated by numeric or ordered categorical features. Further work is needed to extend the ideas to settings with one or more unordered categorical features, such as image classification. In such cases the feature space becomes a zero-volume subset of $\mathbb{R}^d$, and estimating gradients is no longer meaningful. Also, while our data example suggests that RbX may be reasonably robust to the choice of context points, the number of context points needed to adequately cover the space of plausible feature values increases drastically with dimension $d$ due to the curse of dimensionality. The sampling procedure for LIME and the computational complexity of SHAP suffer from similar issues, while the gradient based methods do not but at the expense of only examining $\hat{f}$ in a small number of directions, which we have argued hurts detection power.

Finally, there may also be other ways in which the polytope $\mathcal{P}$ constructed by Algorithm 1 or a variant thereof could be useful for local prediction importance, such as escape distances from $\mathcal{P}$ in directions not parallel to the coordinate axes. We also believe theoretical work on polytopic approximation (Bronstein, 2008; Arya et al., 2012) could be leveraged in future study to develop mathematical guarantees about how well $\mathcal{P}$ approximates $\mathcal{E}$. Any such guarantee would require some prior knowledge about the structure of $\hat{f}$, such as smoothness or local monotonicity; for a completely generic $\hat{f}$, the level sets can be arbitrarily non-convex. Requiring knowledge of such structure would deviate from the strict model-agnostic paradigm, but would be practically useful in a setting where some information about $\hat{f}$ can be made known to the party interested in local prediction importance.

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

## A   APPENDIX

### A.1   LINE SEARCH ALGORITHM

Here we provide a sample line search algorithm for shrinking the context points given to the RbX algorithm to the $\epsilon$-boundary.

---

**Algorithm 3** Line-Search Algorithm

---

1: **Input:** $\epsilon$-far point $x \in \mathbb{R}$ to be shrunk, thresholds $\epsilon \succeq 0$, prediction model $\hat{f}$, target $x_0 \in \mathbb{R}^d$, and maximum number of iterations $M$.
2: Set $t_H \leftarrow 1, t_L \leftarrow 0$.
3: **for** iter $\in [1 : M]$ **do**
4:      Compute $t_M \leftarrow \frac{1}{2}(t_H + t_L)$
5:      **if** $x_0 + t_M(x - x_0) \in \mathcal{E}$ **then**
6:          $t_L \leftarrow \frac{1}{2}(t_H + t_L)$
7:      **else**
8:          $t_H \leftarrow \frac{1}{2}(t_H + t_L)$
9:      **end if**
10: **end for**
11: **return** $\frac{1}{2}(t_H + t_L)$

---

### A.2   TRUSTWORTHY REGIONS

As discussed by Mase et al. (2019), a pitfall of baseline methods such as LIME, Kernel SHAP, and IG is that they often rely on predictions at implausible combinations of feature values, such as a graduation date before a birth date, due to interactions between features. Local prediction explanations utilizing such information have questionable fidelity.

To prevent the feature escape distances $S_j(\mathcal{P})$ from using information about the classifier near implausible feature values, we can establish a "trustworthy region" $\mathcal{T}$ containing $x_0$ that corresponds to the set of plausible feature values. Then if $S_j^+(\mathcal{T}) < S_j^+(\mathcal{P})$ — meaning that in order to escape $\mathcal{P}$ by increasing the $j$-th feature from $x_0$, we must leave the trustworthy region $\mathcal{T}$ — we set $S_j^+(\mathcal{P}) = \infty$. We do the same thing for the $S_j^-(\mathcal{P})$. We could also make the simple feature escape distances $S_j(\mathcal{E})$ more trustworthy in the same way.

In some settings, domain knowledge informs a reasonable choice for $\mathcal{T}$. Otherwise, there are many plausible ways to define $\mathcal{T}$, assuming access to a large collection of plausible feature values, such as the context points for the RbX algorithm. One such method is given in Appendix 5 of Mase et al. (2019). Another, based on Section 14.2.4 of Hastie et al. (2009), would be to estimate $r(x) = \frac{g(x)}{g_0(x)}$ where $g(\cdot)$ is viewed as an unknown joint density of the data generating process for the context points, and $g_0(\cdot)$ is a known baseline density that is positive at each context point, e.g. uniform over a rectangular region containing the context points. This function $r(x)$ can be estimated by any binary classification procedure that outputs class probabilities. By generating a large number of i.i.d. points from $g_0(\cdot)$, we can learn the probability that a point at $x$ came from $g$ rather than $g_0$, using both the original context points and the feature combinations generated from $g_0(\cdot)$. From this, an estimate $\hat{r}(x)$ of $r(x)$ follows via Bayes' rule. Then we define $\mathcal{T} = \{x \mid \hat{r}(x) \geq \beta\}$ for some trustworthiness threshold $\beta > 0$.

We illustrate the latter algorithm using a simple quadratic logistic regression classifier fit to the popular Boston housing dataset (Harrison Jr & Rubinfeld, 1978). The goal, as in Mase et al. (2019), is to predict whether median neighborhood value is less than \$20,000 based on two features — CRIM (crime rate) and RM (median number of rooms per home). We use the same target point as Mase et al. (2019), marked with a red X in Fig. 4.

The trustworthiness classifier was fit with multivariate adaptive regression splines (Friedman, 1991) using a logistic link, allowing for order-two interactions. The contours of this classifier are shown in Fig. 4. If $\mathcal{T}$ is defined with any trustworthiness threshold $\beta$ larger than about 0.15, the escape distance in the positive CRIM direction is set to $\infty$ because then it is impossible to escape the RbX

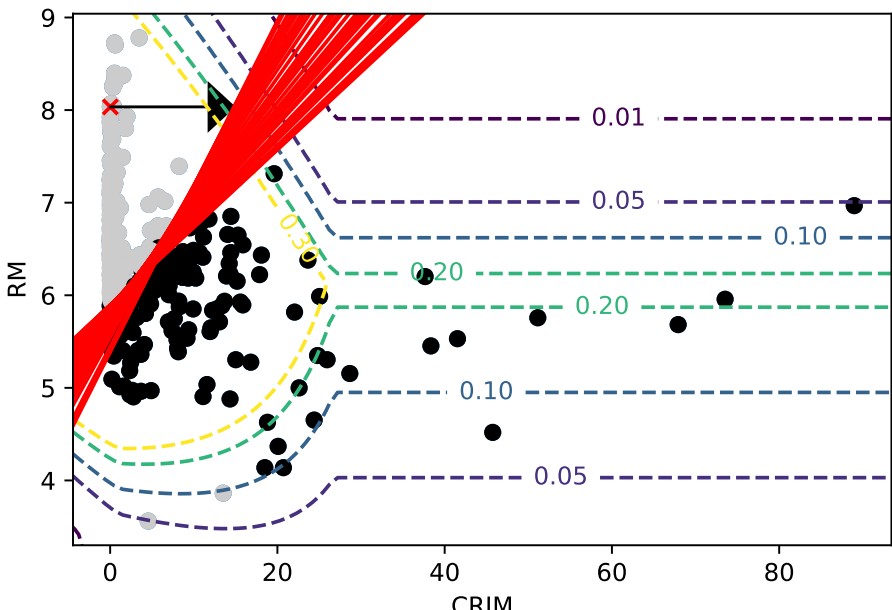

Figure 4: An illustration of the trustworthiness region classifier for the Boston housing data example described in the text. Its contours are indicated by the dashed lines. The solid lines indicate the halfspaces defining the polytope $\mathcal{P}$ from the RbX algorithm for the target point $x_0$, denoted with a red X. The arrow indicates the "polytope escape path" from $x_0$ in the CRIM direction. For any $\beta$ such that the arrow crosses the $\beta$-contour (or smaller) of the trustworthiness region classifier, we set $S^+_{\text{CRIM}}(\mathcal{P})$ to $\infty$. The dots are candidate context points from the entire dataset, colored by whether they are $\epsilon$-far.

polytope without leaving $\mathcal{T}$. As Mase et al. (2019) suggest, this is desirable since there are no context points with similar RM values in the $\epsilon$-far region but high values of CRIM.

### A.3 RESULTS FOR BOOSTED TREE ENSEMBLE ON FICO DATASET

We replicate the results given in Table 1, but replacing the tree classifier in Fig. 2 with a sparse gradient boosted ensemble fit using XGBoost (Chen & Guestrin, 2016) on the same training dataset. Sparisty is enforced by only training on the 5 features that appear in the simple tree classifier of Fig. 2, which are also the first 5 features in Table 1. The training loss is the negative logistic log likelihood with early stopping after 10 iterations (chosen to roughly minimize out of sample classification error). Table A.3 is an analog of Table 1 in the main text, for the same target point $x_0$. Under the ensemble classifier, the predicted probability of "Good" RiskPerformance is 0.241.

Table 2: Same as Table 1, but for the boosted tree ensemble classifier

| Feature | Feature value | $S_j(\mathcal{P})$ | $\tilde{S}_j(\mathcal{P})$ | LIME | SHAP |
|---|---|---|---|---|---|
| ExternalRiskEstimate | 61.0 | 0.48 | 0.062 | -0.12 | -0.21 |
| MSinceOldestTradeOpen | 149.0 | 27.5 | 0.30 | -0.056 | -0.020 |
| MSinceMostRecentInqexcl7Days | 0.0 | 4.16 | 0.97 | -0.074 | 0 |
| MSinceMostRecentDelq | 3.0 | 5.17 | 0.25 | -0.038 | -0.022 |
| AverageMInFile | 49.0 | 41.4 | 1.54 | -0.028 | -0.090 |
| NumTrades60Ever2DerogPubRec | 1 | $\infty$ | $\infty$ | -0.0004 | 0 |

