# OpenReview forum: "RbX: Region-based explanations of prediction models"
_ICLR.cc/2023/Conference — Submitted to ICLR 2023_

### Official Review · Reviewer_2PQz · 2022-10-23

**Confidence:** 4
**Correctness:** 2
**Technical Novelty And Significance:** 3
**Empirical Novelty And Significance:** 2
**Recommendation:** 5

**Clarity, Quality, Novelty And Reproducibility:**

The work is original to the best of my knowledge.

While the work is promising, the quality is not high enough for admission to ICLR in current form, due to the unstated locally monotone assumption described above as well as due to an insufficiently thorough set of experimental results.

Although most of the paper is very clearly written, there is also one very bad typo on line 11 of Algorithm 1. I'm pretty sure the authors intend a vector norm around x_tilde - x_0, i.e. it probably should read || x_tilde - x_0 || ^2.  The ^2 wouldn't make a difference to the minimization but I'm guessing it was intended to be there given the "2" in x_02 as written.  It's also confusing that the notation below the argmin of x_tilde in R doesn't  use the same fancy-script R as is used for the set of context vectors. I'm pretty sure it's supposed to be the same fancy-script R, indicating the set of context vectors. As it's actually written, it took me a while to decipher what was intended....I was wondering if x_tilde and x_02 were scalars rather than vectors and whether the R below the argmin was just the real line rather than the set of context vectors.   I wasted an annoying amount of time on this. I generally want to be supportive of the authors and I encourage them to continue this line of research, but a typo like this at a key line in a key algorithm is not helpful to one's chances of acceptance.

I think there's also a typo at line 14 of the same algorithm, with the letters "int" unnecessarily following the intersection symbol, before the H_k.


**Strength And Weaknesses:**

The method is novel to the best of my knowledge and I think it has a lot of potential. I like the focus on how the output space changes locally and I like the ability of the method to detect complex directions in input space involving interactions between multiple variables.

However, I don't think this research is ready for publication yet.  One serious concern (which could potentially be addressed in a future version of the work) is that the LineSearch bisection algorithm implicitly makes an assumption (which may be unfounded) that the prediction is locally monotone along the line between x_0 and a context point x. Let's take the example desired interval specified at the bottom of page 3 (prediction 13, desired interval between 10 and 20). Let's say the prediction at the initial, unshrunk context point is 21. So you do the bisection and then evaluate the function at halfway between x_0 and x and the prediction at 0.5*(x_0 + x) is now 19. OK, but that doesn't necessarily mean that the prediction is between 13 and 19 (or even 13 and 20) on the entire line between x_0 and 0.5*(x_0 + x). How do we know it doesn't rise to 25 at e.g. 0.25*(x_0 + x) and then fall down to 17 before rising again to 19 at 0.5*(x_0 + x) if we have no knowledge of the fitted function aside from query ability at individual input points?

This problem is addressable, because in general, you do have access to the properties of the fitted function, which tends to be piecewise constant for a tree or K-NN or piecewise linear if using a ReLu deep network or at least otherwise differentiable and capable of anaytic local analysis if a sigmoid-style deep network or something along those lines is used. So RbX as currently presented plus some verification of locally monotone properties would be on much more solid ground and seems achievable. Although the authors might be tempted to do rewrites in the rebuttal which would address this issue, I think it would be wiser to take more time to address the issue and resubmit either next year or in a different venue.

I also think the presented experiments are much too thin. Just one data point on one real-world problem in the main text and then 4 toy problems is not enough I would like to see much more empirical evidence on real world data that the issues with competing methods like SHAP and LIME arise frequently.

It would also help to spend a bit more time on the distinction between local explanations and more global explanations which are nonetheless important but might be forgotten by a purely local explanation.  For instance, suppose a credit applicant makes $90K/year but doesn't have a college degree. The person gets denied for credit and the local explanation is a lack of college degree.  Suppose the fitted function is a tree which denies everyone with a salary below $70K but ignores additional salary above $70K. A strictly local explanation might tell the applicant that the degree is what is needed and the applicant might erroneously get the impression that approval would be achieved with a college degree and a $60K salary. While it's valuable to have a local polytope that RbX creates, it would still be good to clarify that it may not explain non-local effects like these.

**Summary Of The Paper:**

A novel method (RbX) is presented for generating localized explanations of a fitted model output by creating a region in input space (a polytope formed by the intersection of affine halfspaces ) around a given input vector such that outside the polytope, the fitted model's prediction is outside a user-specified interval around the prediction at the input vector. The method only requires query access to the fitted model, i.e., it only depends on output values at a finite user-specified set of points in input space. RbX does not require a reference/baseline input vector, unlike some competing explanation methods. RbX is also able to detect directions in input space involving simultaneous changes to multiple input variables which achieve classification output changes in cases where a single change along the coordinate of any individual input variable would not achieve a classification output change. The method is illustrated on a single input vector for a decision tree on one real-world credit scoring problem as well as on 4 toy problems. A gradient boosted tree ensemble is also examined in an appendix.

**Summary Of The Review:**

The paper presents a promising idea for model explanations by creating a local region around an input vector where the prediction doesn't change much, but it makes an unstated and unjustified assumption of locally monotone predictions which needs to be examined or defended in future versions of the work and it also does not have sufficient experimental results to justify acceptance. There is also a very confusing typo in the presentation of a key algorithm.

** UPDATE post-rebuttal **

Given the typo fixes and additional discussion of local monotonicity, I have raised my score to a 5.

---

> ### Author Response · Authors · 2022-11-18
> **Reply to Official Review of Paper1425 by Reviewer 2PQz**
>
> We appreciate the reviewer’s careful commentary.
>
> Their main constructive feedback relates to the fact that any given shrunk context point $x_i$ found in the initial step of the RbX algorithm may not actually lie on the closest part of the epsilon-boundary to $x_0$ along the ray from $x_0$ to $x_i$, due to the possible failure of local monotonicity. In other words, there might exist a point on the ray between $x_0$ and the shrunk $x_i$ found by the line search that is on the epsilon-boundary and closer to $x_0$ than $x_i$.
>
> We have added a mention of this issue to Section 3.1 and describe how it is mitigated by having a sufficient set of context points to cover the range of plausible feature values. We recall that RbX is a greedy algorithm, which means that the closest shrunken context points are considered first. Thus, in the example provided by the reviewer, if there were a context point $y$ between $x_0$ and $0.25*(x_0+x)$ (and local monotonicity holds over that shorter line segment), our line search algorithm would correctly shrink y to the closest point on the epsilon-boundary along the ray from $x_0$ to $y$. Then the RbX algorithm would fit a hyperplane approximating the epsilon-boundary at the shrunk $y$, after which it would ignore the halfspace containing the shrunk $x$ (so the part of the epsilon boundary surrounding shrunk $x$ wouldn’t be considered at all).
>
> The reason we need only worry about context points covering plausible feature values is that we believe, as do Mase et al. (2019), that it is not useful to consider the output of $\hat{f}$ at feature values not considered plausible (see Appendix A.2). Of course, in higher dimensions it is more difficult to “cover” the range of reasonable feature values with a reasonable number of context points. We mention this in Section 6, noting that such curse of dimensionality is unfortunately also a feature of existing methods.
>
> Finally, we agree that having knowledge of specific structure about the prediction model might allow us to make stronger statements about local monotonicity and the correctness of the polytope $\mathcal{P}$ in approximating the epsilon-close region $\mathcal{E}$. However, our focus in this work was on fully model-agnostic methods, along the lines of LIME and Kernel SHAP. We’ve indicated this as a potential line of future work in Section 6.
>
> We would appreciate if the reviewer could explain more about the distinction between local and global explanations in the context of their example. We are not sure of the sense in which their example would necessitate a “global” explanation.
>
> Finally, in response to the suggestion for more empirical evidence, we’ve also expanded the scope of our real data example in Section 5 to show the prevalence of the issues with LIME and SHAP described (sparsity, choice of baseline).
>
> Typos:
>
> We apologize for the typos on line 11 of Algorithm 1. The lack of a vector norm appears to have been an issue with PDF conversion as it appeared on our working version; we’ve made sure it is corrected in the rebuttal. The issue about the script R has also been corrected in the rebuttal.
>
> The “int” on line 14 of Algorithm 1 is not a typo, it refers to “interior.” We allude to this usage in the paragraphs below.

---

> > ### Comment · Reviewer_2PQz · 2022-12-06
> > **Thanks for the rebuttal**
> >
> > I have raised by score to a 5, given the additional discussion of monotonicity issues and the typo fixes.
> >
> > I apologize for the weird formatting in my review when discussing global vs. local explanations.  The point I was trying to make is that a local explanation which focuses on one feature might give the impression that other features don't matter at all.  This is a minor concern, however, so I wouldn't worry about it as much as the other issues I raised when continuing this line of research in the future.

---

### Official Review · Reviewer_EFoo · 2022-10-24

**Confidence:** 4
**Correctness:** 3
**Technical Novelty And Significance:** 2
**Empirical Novelty And Significance:** 2
**Recommendation:** 6

**Clarity, Quality, Novelty And Reproducibility:**

The paper is relatively clear in terms of the model and intuition, although there are some areas that need polishing and better understanding of the performance (most notably, conditions that make RbX desirable to be used over other alternatives) require further evaluation. The quality of the ideas are interesting and the model is intriguing. The novelty is more with respect to the application of existing tools to solve a problem in feature relevance of black-box predictions.

**Strength And Weaknesses:**

With respect to its strengths, the paper is well organized, the problem is relevant to the ML community, particularly to model interpretability and black-box model analysis. The paper states a clear goals although the difference of "local prediction importance" vs feature importance assessment is rather small and due to the black-box nature of loca prediction importance. The model is geometrically intuitive which could make it appealing to a wider community. During evaluation the paper  includes a wide range of data generating scenarios including non-linear which would provide support to its validity.
There are some areas where the paper could be improved. For instance, the evaluations seem promising but not fully convincing. The improvement of performance w.r.t the baselines on synthetic data seems small (or none for a few data generating cases such as XOR, Orange skin).  Another weakness is that no evaluation is provided with respect to the choice of parameters e.g. the neighborhood size $\epsilon$. How does the method decay as the number of dimensions grow is not totally clear.

**Summary Of The Paper:**

This paper presents a method (RbX) for generating local feature importance explanations for given predictions (scalar outputs) of a black-box model as a convex polytope that encloses a region (specified by a user) that is close to a target predicted point. The polytope is used to determined the boundary of features that are relevant vs. those that are not (sparsity). Thus, is is some sort of model-agnostic feature importance method, and the authors highlighted this difference with other feature importance methods. The algorithm consists on using a closeness threshold $\epsilon$ to determine a boundary that is build by "shrinking" some context sample points along the line segment joining them with the boundary, i.e. by constructing the segment, finding the point in the boundary closest to the point of interest, and discarding any point outside this boundary. In addition to the boundary RbX needs the gradients of the region which are obtained using finite differences. These are used to compute the importances as scape distances for each feature. The model is evaluated with both real and synthetic data experiments. The paper uses a credit score dataset example (that predicts bad or good riskperformance) to exemplify how RbX Scape Distances compare to scores from LIME and Shap (baselines) in terms of possible interpretation. The synthetic experiments Evaluate 4 data generating scenarios to compare the detection power of local importance methods.

**Summary Of The Review:**

The paper has its pros, such as the relevance, the clear stated goals and the use of non-linear data importance generation. However, there are several reservations with respect to the applicability as the performance is comparable to existing methods. The advantage seems to be more due to the applicability across scenarios (non-linear additive, XOR, etc) but more evaluations are highly recommended. I detailed my comments on what could be improved in the Strength and Weaknesses section.

%%Post rebuttal comments%%
Thank you to the authors for the reply. I think the clarifications, evaluations, and discussions are important additions to the paper. Thus, I have increased my score.

---

> ### Author Response · Authors · 2022-11-18
> **Reply to Official Review of Paper1425 by Reviewer EFoo**
>
> We’d like to thank the reviewer for their constructive feedback.
>
> We highlight that we have added evaluations in Section 5 of our rebuttal revision to determine the sensitivity of RbX to choice of context points, which we believe to be the main tuning parameter for the algorithm. While the reviewer cited epsilon as another possible parameter for which to evaluate the sensitivity of our method, we have tried to emphasize in our revision (particularly Section 2.1) that we view the choice of epsilon as guided by context rather than as a hyperparameter to be tuned.
>
> We have also discussed some limitations of RbX in Section 6, specifically addressing the reviewer’s point about how the performance might decay as the feature dimension grows. We suggest that the variability of our procedure does unfortunately suffer from the “curse of dimensionality” from lack of adequate coverage of a reasonable number of context samples. But we believe this is far better than the widely used existing methods (LIME, SHAP, IG) that only examine the predictions in a small number of directions, on the order of the feature dimension.

---

### Official Review · Reviewer_UFKv · 2022-10-25

**Confidence:** 3
**Correctness:** 3
**Technical Novelty And Significance:** 2
**Empirical Novelty And Significance:** 2
**Recommendation:** 5

**Clarity, Quality, Novelty And Reproducibility:**

Clarity: While each sentence is more or less clear, I found the paper very difficult to follow overall.
- In the introduction, the discussion of local prediction importance and local feature importance is confusing. In particular, when describing feature importance you write, “these approaches fix the prediction model f, but provide importance measures based on changes in predictive performance of that model”; isn’t this similar to your approach in that you’re characterizing how much change is required in certain features to change predictions? Also, it does not seem critical to the exposition–at least at this stage. If your goal is to distinguish your work from others, perhaps this would be better explained in the previous work section.
- In Section 2, can you explain the statement about distances in directions parallel to the coordinate axes informing local sensitivities of f? Specifically, why only in directions parallel to the coordinate axes? Also, please include more detail regarding what you mean by local sensitivities and “desirable” in that sentence.
- In Section 2.2 you make an argument for testing with sparse models. Can you explain this better, especially in contrast with dense prediction models? If this is a characteristic or assumption of your setup, please clarify this earlier–perhaps in the introduction.
- In Section 2.3, you argue that specifying a region of prediction values on the outcome scale is simpler and more interpretable. Can you give evidence and/or examples to support this claim?
- When the user specifies a closeness region, what is the dimensionality of \epsilon that hey need to specify?
- In section 2 and 3 you mean “feature combinations” which appear in related literature. Please define this or give an example.
- In the description of the algorithm is “shrinking” the same as projecting the context points? Can you give more intuition about why context points require shrinking before we find the closest to the target point, i.e., why not just find the closest point
- In Algorithm 1, line 11: what is the subscript 2 for?
- The toy example, especially the visualization, is quite helpful.
- In section 4, the concept of untrustworthy and unlikely parts of feature space are used with no introduction. How are these defined/discovered? Additionally, at the end of the Section you use the notation S_1^tilda, which I don’t think has been defined. This makes the 2nd to last sentence of this section difficult to understand. Perhaps label the figure to add the example you refer to.
- In the synthetic data experiments, my understanding is that you are testing 4 different data generating processes, is this correct? I found this confusing, especially because of the textual description of feature switching, which initially made me think that 1 of the 4 processes was invoked depending on the raw values sampled from the normals. Also, please define the “:=” notation used.
- What are globally/locally relevant features? Please define this in your description of the model. What is the difference between escape distances on the original scale of the features and “simple feature escape distances” as discussed in the real experiment?
- Add y-axis label to Figure 3.
- It’s difficult to know which methods work best simply by reading Table 1. Perhaps add additional details in the caption or annotations in the table to explain the table better.

Quality:
- I found it difficult to get a detailed understanding of the motivation and method from this paper.
- The algorithm proposed seems to be technically sound, barring any missed details from clarity issues.
- The experiments leave much to be desired. It is difficult for me to evaluate how practical the method is: I’m unsure of whether the method can be used with any model, what problems it is not well-suited for, or how much of a burden this method places on the user.
- Additionally, experimental results on the real data are reported on a single test case, which raises questions of robustness.
Some of these issues might be remedied by further clarity in the paper and an explicit limitations section.

Novelty:
- The proposed algorithm is simple, yet novel.
- The idea of explain black box predictions via convex polytopes is new, yet it borrows many ideas from a variety of previous work: explaining points via regions is similar in spirit to hierarchical decompositions of a point set, e.g., with k-d trees, r-trees, ball-trees, etc., and the idea of using distance to a decision boundary for explanation has similarities to work on recourse is algorithmic decision making.

Reproducibility: The algorithm presented does not seem very difficult to implement, but the lack of clarity with respect to details makes me doubt the ease of reproducibility. Additionally, no code was submitted.


**Strength And Weaknesses:**

Strengths:
- The method only requires query access to a black box model, like LIME or SHAP. This is useful for explanation methods.
- The method is simple, intuitive, and reasonable.
- The proposed method satisfies the sparsity property.
- In experiments, especially synthetic, the proposed method recovers the most relevant features consistently.

Weaknesses
- The method is only applicable for prediction problems with a scalar output. This seems limiting.
- The method requires a user to specify a region in output space that is close to a test point. No evidence is given that this is a simple or intuitive thing for a user to do in most cases. While it is possible for a user to specify a single set of epsilon parameters for all test points, it’s conceivable that specific test points merit specific epsilons. While this flexibility is possible, it also burdens the user of the method. (**Update after Rebuttal**: the paper now includes some examples of how setting \epsilon could be intuitive.)
- Empirical study should be improved. There is only 1 experiment with real data and one with synthetic data. In the real experiment, only the explanation of 1 point is evaluated.
- Experiments are performed with random forest, bayes, and knn classifiers. No deep models are used, which calls into question the efficacy of the proposed method with these models. To be clear: it is not necessary for the work to be about deep models, but some discussion of whether or not the method is likely to work with these models is necessary given their prevalence. Intuitively, it seems like a convex polytope approximation of the closeness region may not be effective with highly non-convex models.
- No explicit discussion of the limitations of the method; consider adding a limitations section. (**Update after rebuttal**: the revision includes some discussion of limitations, but the discussion should be expanded).
- Many details are unclear; see below.


**Summary Of The Paper:**

This paper introduces a method for explaining any prediction of a black box model by constructing a convex polytopes that surrounds the prediction, such that leaving the polytope changes the prediction, in some sense of closeness. The proposed algorithm works by iteratively finding the closest “shrunken” point to a test point, estimating the gradient at that point, and using the gradient to compute a corresponding halfspace. The combination of halfspaces are used to construct the polytope, and distances to the edge of the polytope help to characterize how changing certain features affects predictions. The method is employed in two experiments, one with real data and one with synthetic data. In the real data experiment, the method proves to be more consistent than SHAP and better able to identify unimportant features than LIME. In the synthetic experiments, the proposed method is able to identify relevant features better than other competing algorithms.


**Summary Of The Review:**

The method put forth in this paper seems reasonable but understanding its details was difficult due to exposition. Experiments were run with only a handful of simple models and 2 datasets, making it difficult to gauge the utility of the method in practice and its limitations.

---

> ### Author Response · Authors · 2022-11-18
> **Reply to Official Review of Paper1425 by Reviewer UFKv (1/2)**
>
> We appreciate the extremely detailed commentary from this reviewer on our paper. We have tried our best to incorporate their suggestions into our rebuttal revision.
>
> As suggested by this reviewer and others, we’ve included additional evaluations in Section 5 of our revised paper, and provided a more explicit discussion of some limitations in Section 6 of the paper. The additional evaluations concern the stability of RbX to the choice of context points, compared to analogous sources of variability in SHAP (choice of baseline) and LIME (sampling procedure).
>
> One key point raised by this reviewer concerned the choice of epsilon in our algorithm. In our revision, we have clarified that we strongly believe the choice of epsilon places a smaller burden on the user than similar decisions, such as choice of baseline features, required by many of the common algorithms in use (SHAP, IG, etc.). We have provided several examples of epsilon in Section 2.1 of our revision aimed at better illustrating this point.

---

> > ### Author Response · Authors · 2022-11-18
> > **Reply to Official Review of Paper1425 by Reviewer UFKv (2/2)**
> >
> > This reviewer also had numerous concerns around exposition and clarity. Here is our attempt to address each of the specific concerns that the reviewer laid out under “clarity”:
> > - We have clarified that the methods of Fisher et al. (2019) and Casalicchio et al. (2018) derive feature importance from changes in predictive performance, whereas our method looks at changes in the actual values of the predictions themselves, without regard to how well these predictions approximate “reality” (i.e. predictive performance). The goal of introducing this discussion early in the paper is to clearly state the problem we are solving.
> > - Our later results suggest that directions parallel to the coordinate axes for escaping the polytope (as opposed to “simple feature importance”, which only seeks to escape the original epsilon-close region) is sufficient to yield good performance over competing methods. Of course, other directions might work even better and could be worth exploring, as now discussed in Section 6. By “desirability” we refer to our two axioms, sparsity and detection power. We have reworded to make these points clearer.
> > - The argument for sparsity is simply that if a given feature does not appear at all in a prediction model, it should not be assigned any importance, which we believe should not be very controversial (although we show LIME does not satisfy this). We’ve added a sentence indicating this, as well as clarifying that our goal in developing RbX is to maximize detection power subject to sparsity. Our procedure does not require the prediction model to be sparse, but if it is sparse, we don’t want irrelevant features assigned importance. The focus on sparse prediction models for evaluation is to ensure the ground truth is clear. We have included an additional citation to a paper by Zhou et al. (2022) suggested by another reviewer which also believes in evaluation criteria similar to sparsity and detection power, though they perform the evaluations in a different way.
> > - As discussed above, we have added several examples to Section 2.1 of our revision describing how to choose epsilon, hopefully demonstrating that the choice is quite natural depending on the context.
> > - For a scalar prediction model, $\epsilon = [\epsilon_L, \epsilon_H]$ has dimension 2, which is hopefully clearer in the revision.
> > - By “feature combinations” we simply meant values of features. We have avoided this terminology in our revision.
> > - We need to shrink context points to the epsilon decision boundary because we need to approximate the epsilon decision boundary. The unshrunk context points are not, in general, on the decision boundary.
> > - Algorithm 1 line 11: This was an unfortunate typo pointed out by another reviewer, it should refer to the Euclidean norm of the vector $\tilde{x}-x_0$.
> > - We are glad the reviewer found the toy example helpful.
> > - We’ve clarified that figuring out the plausibility of a feature combination is explained in our Appendix. Trustworthiness is defined in the Appendix and no longer referred to in the main text, to avoid confusion from readers who do not look at the Appendix. We apologize for the inconsistent use of the tilde notation in our original draft. We have made the notation in this section consistent so that $S_j(\mathcal{P})$ refers to the RbX escape distances on the scale of the original features (looking at the polytope P), $\tilde{S}_j(\mathcal{P})$ is the same but on the standardized feature scale, while $S_j(\mathcal{E})$ is a simple feature escape distance which ignores the polytope P altogether.
> > - The experiments indeed consider 4 different data generating processes. We have clarified this in our revision. The := notation meant “defined to be equal”; we have removed this notation from our revision for clarity.
> > - We’ve clarified what we mean by the locally and globally relevant features in our synthetic experiments. As defined in Section 4 (and hopefully clarified by the corrections to the notation described above), the escape distances on the original scale of the features are the $S_j(\mathcal{P})$, i.e. they use the RbX polytope and are on the original scale of the features. The simple feature escape distances are $S_j(\mathcal{E})$, which do not use the polytope constructed by RbX.
> > - We’ve added y-axis labels to Fig. 3.
> > - We’ve clarified in the caption of Table 1 that the RbX escape distances are on the feature scale, compared to the other methods (LIME and SHAP) which give importance scores on the outcome scale.

---

> > > ### Comment · Reviewer_UFKv · 2022-12-12
> > > **Informative response and revision**
> > >
> > > Thank you for the detailed responses and also for the provided revision. I think that the new discussion of the \epsilon parameter is helpful, and for this I will increase my score. Adding the limitations to section 6 improves the paper, but I still think that further discussion of limitations is merited. Finally, additional analysis in the experiments section is a good start, but ultimately, in my opinion, this method's utility must be proven out via additional empirical experimentation, i.e., datasets, baseline models, and analysis.

---

### Official Review · Reviewer_4Xhj · 2022-10-25

**Confidence:** 4
**Clarity, Quality, Novelty And Reproducibility:** The paper is well written, novel, and…
**Correctness:** 4
**Technical Novelty And Significance:** 3
**Empirical Novelty And Significance:** 3
**Recommendation:** 6

**Strength And Weaknesses:**

strengths

- The proposed technical idea of region based explanations is quite novel and useful. The idea of using escape distances is good.
- Although the authors do not mention this or show via empirical results, the method might even have other indirect benefits such as the variance of explanations across similar samples might be low (needs to be checked), which would make this method a strong contender compared to LIME.

weakness

- Although the idea is good, the more evaluations are needed. The synthetic experiments are good. However it would be good to evaluate the method using several explainability metrics already available in the literature that do not require ground truth explanations and some that proposes to test against ground truth (https://arxiv.org/pdf/2104.14403.pdf)

- Potential limitations of this method are not fully evident from the experiments.


**Summary Of The Paper:**

A new region based explanation method is proposed that computes attribution scores for features as "escape distances" from a region/polytope defined by the user as input.

**Summary Of The Review:**

See above.

---

> ### Author Response · Authors · 2022-11-18
> **Reply to Official Review of Paper1425 by Reviewer 4Xhj**
>
> We appreciate the reviewer’s comments on our paper.
>
> To address the recommendations of this reviewer and several others about including more substantive evaluations, in Section 5 of our rebuttal revision we have added some evaluations concerning the sensitivity of the ordering of the feature scores found by SHAP, RbX, and LIME to various inputs. These inputs include choice of baseline in SHAP, choice of context points for RbX, and the sampling procedure for LIME. The evaluations, conducted over a range of different target points, suggest that for the credit scoring example, the explanations from RbX are less variable over choices of context points than the explanations from SHAP over baselines varying with the same noise level. They also support this reviewer’s hypothesis that the variance of RbX’s explanations across context samples is quite a bit lower than that of LIME.
>
> We have also added some discussion of limitations to Section 6 of the revision. This was also suggested by multiple other reviewers.
>
> Finally, we would also like to thank the reviewer for bringing our attention to the Zhou et al. (2022) paper. We are glad to see that that paper fundamentally supports criteria like sparsity and detection power as a “ground truth.” We cite it at the beginning of Section 5 in our rebuttal revision and have clarified throughout the revision that we choose to evaluate our methods on sparse prediction models. The difference is that Zhou et al. (2022) considers more complex, non-sparse black-box models where the ground truth is harder to determine. By contrast, we consider (locally) sparse prediction models throughout our evaluations, so that the ground truth is explicit and does not require constructing the synthetic datasets of Zhou et al. (2022).

---

### Decision · Program_Chairs · 2023-01-20

**Decision:**

Reject

**Justification For Why Not Higher Score:**

Below borderline.

**Justification For Why Not Lower Score:**

N/A

**Metareview: Summary, Strengths And Weaknesses:**

The paper introduces a new region-based black box explanation method that computes attribution scores for features as escape distances from a region/polytope defined by the user as input. The geometry of this polytope can be used to quantify the local sensitivity. Some empirical evaluation is also provided; and further analysis was done in the discussion phase. While the reviewers agree that the paper is interesting, further empirical analysis is required to move the paper beyond borderline.